# Pneumatic Retinopexy Involving the Use of Intravitreal Air Injection and Laser Photocoagulation for Rhegmatogenous Retinal Detachment in Phakic Eyes

**DOI:** 10.3390/jpm13020328

**Published:** 2023-02-14

**Authors:** Tiepei Zhu, Zhenyang Xiang, Qinzhu Huang, Gaochun Li, Shenchao Guo, Enhui Li

**Affiliations:** 1Eye Center, Second Affiliated Hospital of Medical College, Zhejiang University, Hangzhou 310058, China; 2Department of Ophthalmology, Taizhou Hospital of Zhejiang Province, Taizhou 318000, China

**Keywords:** retinal detachment, pneumatic retinopexy, intravitreal injection, laser photocoagulation, retinal thickness

## Abstract

The clinical efficacy of pneumatic retinopexy (PR) using intravitreal pure air injection and laser photocoagulation for rhegmatogenous retinal detachment (RRD) remains unknown. Thirty-nine consecutive patients with RRD (39 eyes) were included in this prospective case series. All patients underwent two-step PR surgery containing pure air intravitreal injection and laser photocoagulation retinopexy during hospitalization. The main outcomes were best-corrected visual acuity (BCVA) and primary anatomic success rates after PR treatment. The mean follow-up was 18.3 ± 9.7 months, ranging from 6 to 37 months. The primary anatomic success rate was 89.7% (35/39) after PR treatment. Final reattachment of the retina was achieved in 100% of cases. Macular epiretinal membrane was developed in two patients (5.7%) among successful PR cases during the follow-up. The mean logMAR BCVA value was significantly improved from 0.94 ± 0.69 before surgery to 0.39 ± 0.41 after surgery. The average central retinal thickness was significantly thinner in the RRD eyes of macula-off patients (206.8 ± 56.13 μm) when compared with the fellow eyes (234.6 ± 48.4 μm) at the last follow-up (*p* = 0.005). This study concluded that an inpatient PR procedure with pure air injection and laser photocoagulation is a safe and effective approach to treating patients with RRD, who may achieve a high single-operation success rate and good visual acuity recovery.

## 1. Introduction

Rhegmatogenous retinal detachment (RRD) is potentially an acute vision-threatening disease and requires surgical intervention as early as possible. Surgical approaches commonly include scleral buckling (SB), pars plana vitrectomy (PPV), pneumatic retinopexy (PR), or combinations of these treatments [1]. Although anatomic reattachment is successful in most cases, optimal management for a vast number of RRD cases remains controversial.

PR is considered the most minimally invasive treatment for repairing simple RRD. This office-based surgery has been well accepted by retinal surgeons as an alternative option for simple RRD with superior retinal breaks [2]. A recent multicenter, randomized, controlled clinical trial was performed to compare the outcomes of PR to PPV for primary RRD, the results of which suggested that PR should be considered the first-line treatment for primary RRD meeting the study criteria [3].

The successful anatomical retinal reattachment rate of single PR treatment ranged from 50% to 90% in previous studies [3,4,5,6,7,8,9]. The PR procedure can be performed in one step, with the application of scleral cryopexy to retinal breaks just before intravitreal gas injection, or in a two-step manner, with initial intravitreal gas injection to flatten the detached retina, followed by either laser photocoagulation or scleral cryopexy to seal the retinal break [2]. Both procedures have been applied in previous studies, depending on the surgeon’s discretion [3,5]. Thus, the efficacy of PR for simple RRD may be compounded due to the different procedures used in a study.

Scleral cryopexy has been mainly applied to retinopexy in previous PR studies [3,4,5,10], and expansile gas such as sulfur hexafluoride (SF6) or perfluoropropane (C_3_F_8_) was then injected into the vitreous to ensure the internal tamponade of the retinal breaks [3,4,5]. However, both scleral cryopexy and injection of the expansile gas may have a higher incidence of proliferative vitreoretinopathy [11,12]. Moreover, expansile gases may be unavailable in some healthcare settings. To date, the clinical outcome of PR via laser retinopexy and pure air injection for RRD is still not well characterized.

This study aims to explore the efficacy of two-step PR in which pure air intravitreal injection and laser retinopexy are used in a consecutive series of patients with RRD. It has been reported that successful retinal reattachment can be achieved after as little as 6–8 h of gas tamponade, with an appropriate head position [13]. Thus, unlike surgeons performing traditional office-based procedures, we performed the PR treatment as an inpatient procedure to (1) maintain the appropriate head position after gas injection and adjust the position in a timely manner when the retinal break was not blocked and (2) apply laser retinopexy as early as possible after the retina was attached.

## 2. Materials and Methods

### 2.1. Design and Patients

This was a prospective case series study of consecutive patients with RRD who visited the Department of Ophthalmology of Taizhou Hospital between 1 April 2019 and 31 March 2022. The inclusion criteria were as follows: had primary RRD and had not received prior treatment; presented with one break or multiple breaks separated by less than 3 clock hour positions; phakic eye; presented superior breaks located at the 8 to 4 o’clock position; presented no or minimal PVR (grade B). The exclusion criteria were as follows: retinal break size larger than one clock hour; PVR grade C or D, or retinal star folds exerted traction on the breaks; presented retinal breaks in the inferior 4 clock hour positions of the fundus; presented a giant retinal tear or dialyses; had severe or uncontrolled glaucoma; presented cloudy media such as cataracts or vitreous hemorrhage that precluded an adequate view of the peripheral retina; and had a history of prior retinal surgeries. Patients with a physical disability or mental incompetence who could not maintain the required head position were also excluded. 

All the participants included in our study underwent PR surgery containing pure air intravitreal injection and laser retinopexy. The possible benefits and risks of PR treatment were explained to the patients and informed consent was obtained from all adult patients and from the parents if the patient was younger than 18 years old, in accordance with the Helsinki Declaration, before inclusion in the study. Institutional review board approval (X20180301) for this study was obtained from the Ethics Committee of Taizhou Hospital.

### 2.2. Surgical Technique

Careful peripheral retinal examination with scleral depression was applied to identify any pathologic features in each patient. Laser retinopexy was performed if any lattice degeneration or retinal breaks existed in the attached retina before PR treatment. The PR procedure involves an intravitreal pure air injection first, followed by laser retinopexy when the retina is attached. The air injection was performed under topical anesthesia in the operating room. For the anterior chamber paracentesis, gentle pressure was applied with a cotton-tipped applicator to facilitate more aqueous drainage. Then, 0.6–0.8 mL of filtered sterile air was injected 3.5 mm posterior to the limbus, depending on the extent of the retinal breaks and the eye size. The usual injected gas volume was 0.6 mL; however, 0.6–0.8 mL was injected if the patient had one retinal break size larger than 2 papillary diameter (PD) or multiple breaks separated by more than 1 clock hour or high myopia; a larger air bubble or a longer duration of tamponade was required in those specific patients. Then, the placement of a gas bubble over the retinal break(s) and the perfusion of the central retinal artery were confirmed with indirect ophthalmoscopy. Anterior chamber paracentesis was repeated in the situation of marked intraocular pressure elevations. All the patients were asked to maintain a proper head position continuously during hospitalization. Retinal laser photocoagulation was administered as early as 6–8 h after air injection and a supplemental laser was applied within 1–3 days during residency. The head was tilted when the residual gas bubble obstructed the laser pathway. Usually, 3–4 rows of laser spots were made around the retinal break. All the patients were followed up at 1 week, 1 month, 3 months, and 6 months post operation, then every 6 months thereafter.

### 2.3. Clinical Measurements

Each patient in the study underwent a complete ophthalmologic assessment, including slit-lamp examination, fundus examination with indirect ophthalmoscopy under mydriasis, and intraocular pressure measurements. Before PR surgery, clinical data—which include sex, age, lens state, the most recent refraction data before retinal detachment, best-corrected visual acuity (BCVA), days to PR treatment, the extent of the RRD area, and the number of breaks—were recorded. The main surgical outcomes were postoperative BCVA and primary PR anatomic success rates, which were defined as the ratio of successful retinal attachment after PR treatment within 6 months post operation. PR failure was defined as the need for additional intervention (scleral buckle and/or vitrectomy) for retinal detachment after initial PR treatment. In addition, macular status was confirmed by optical coherence tomography (OCT) before and after PR surgery.

### 2.4. Statistical Analysis

For statistical analysis of the data, we used R software for Windows (version 3.4.3; http://www.R-project.org). The Shapiro–Wilk test was applied to assess the normal distribution of the data. Comparison of data between groups was performed using one-way analysis of variance when samples were normally distributed or the Kruskal–Wallis test when parametric statistics were not possible. The potential effects of preoperative baseline characteristics on surgical outcomes were analyzed by linear mixed-effects models using the “lem4” package. *p* values below 0.05 were considered statistically significant.

## 3. Results

A total of 39 eyes of 39 patients with RRD met the inclusion criteria; these patients were enrolled in this study. The mean age of the included patients was 51.8 ± 14.5 years (range: 14–75 years). All the patients had phakic eyes and no history of intraocular surgery. The preoperative characteristics of all patients are summarized in Table 1. Twenty-one patients (21/39) were evaluated for a grade A level of proliferative vitreoretinopathy (PVR) before surgery, and the remaining cases had PVR with a grade B level. The size of retinal breaks ranged from 0.5–4 PD. Most of the breaks were typical horseshoe-shaped tears (79.4%, 31/39), and the remaining cases were identified as round-shaped holes (20.6%, 8/39). There was no significant difference in terms of age, sex, days to surgery, or number of retinal breaks between macula-on and macula-off patients. However, there was worse visual acuity and more quadrants of detached retina in the macula-off group versus the macula-on group (*p* < 0.05).

The average follow-up was 18.3 ± 9.7 months, ranging from 6 to 37 months. The primary anatomic success rate was 89.7% (35/39) after PR treatment at the last follow-up (Figure 1), and the final reattachment rate was 100% (39/39) when combined scleral buckling or PPV surgery was applied for the failed PR cases. Among the 35 patients who had successful PR cases, 34 patients (97.1%) received only a single air injection, and 1 patient needed an additional air injection because the retina was partially attached, but the air was almost completely absorbed. For postoperative complications, one patient developed a new retinal break superiorly and shallow retinal detachment two days after the initial injection. The head position was adjusted in a timely manner to close the new break with the residual air bubble, and the retina was finally attached after supplemental laser treatment. Two patients (5.7%) with successful PR cases developed macular epiretinal membranes during the follow-up. One successful PR case developed steroid-induced glaucoma at the last follow-up, which was not included for further analysis. The four failed PR cases were all macula-off type retinal detachment, and the reasons for failure were gas bubbles under the retina (one case), missed retinal break (one case), and retinal break unblocking (two cases). All of these patients underwent additional retinal repair surgeries (PPV for three patients and scleral buckling for one patient) within one week after initial PR treatment and finally had retinal attachment during the follow-up.

Postoperative best-corrected logMAR visual acuity significantly improved after retinal detachment repair surgery (Figure 2). Among successful PR cases, the preoperative BCVA of logMAR in the macula-on group was 0.37 ± 0.28 and improved to 0.16 ± 0.17 after PR treatment (*p* < 0.001, *n* = 20). The macula-off group also showed significant improvement in postoperative BCVA (1.62 ± 0.36) of logMAR over baseline (0.59 ± 0.48, *p* < 0.001, *n* = 14). The linear mixed model demonstrated that the baseline BCVA had a negative effect on the changes in BCVA, which means that the worse the baseline BCVA was, the greater the improvement achieved after surgery. In the macula-on group, no significant difference in average central retinal thickness was observed between the baseline value (236.6 ± 31.3 μm) and the postoperative eye value (231.4 ± 31.1 μm, *p* = 0.589, *n* = 16). In the macula-off group, central retinal thickness was significantly thinner in the RRD eyes (206.8 ± 56.13 μm) compared with the fellow eyes (234.6 ± 48.4 μm) at the last follow-up (*p* = 0.005, *n* = 13).

## 4. Discussion

Many variations in PR treatment have been described in previous studies [2,3,4,5,7,8]. Different gas types—including C_3_F_8_, SF_6_, and pure air—can be injected, and different retinopexy approaches, such as cryopexy and laser photocoagulation, have been applied before and after gas injection. In this study, we performed a two-step PR procedure for RRD patients and further evaluated the efficacy of pure air injection followed by laser retinopexy to treat the detached retina. The outcomes demonstrated a high success rate of retinal attachment, particularly in macula-on RRD cases.

SF_6_ and C_3_F_8_ are the gases most frequently used in the PR procedure. Because of the low solubility of these expansile gases, the average durations of SF_6_ and C_3_F_8_ were 12 days and 38 days, respectively [10]. The prolonged longevity of expansile gas bubbles may have disadvantages. Patients need to maintain their head position for a longer period, and air travel is also prohibited. Long-lasting gas bubbles can increase the risk of cataract formation in phakic eyes [14]. Furthermore, the movement of bubbles may produce extra vitreo-retinal traction and induce new retinal breaks. Thus, nonexpansile and filtered air could be used for retinal fixation. It has been suggested that laser photocoagulation could produce mild chorioretinal adhesion within 24 h, moderate adhesion after approximately 5 to 7 days, and firm adhesion after 2 weeks [15]. A 0.8 mL intravitreal air bubble has an average duration of 4 days, which is probably sufficient time for most cases to form a moderate chorioretinal adhesion. Sinawat et al. [15] demonstrated that PR in which filtered air was used had a similar initial reattachment rate compared with that of C_3_F_8_ injection in a randomized noninferiority trial, although the approach of retinopexy was not mentioned in the study. Yee et al. [16] reported that office-based PR in which only pure air was used yielded a high success rate and long-term efficacy. Similarly, our study used approximately 0.6–0.8 mL of filtered air to cover the detached retina up to 120 degrees and block the retinal breaks. Unlike the expansile gas, the larger pure air bubble could reach the maximum amount immediately after initial injection and had higher buoyancy and surface tension at the early stage of PR treatment, which may facilitate epithelial pump removal of the subretinal fluid. Therefore, this initial large air bubble could allow us to treat inferior retinal detachment cases with multiple breaks separated within 2 h. Additionally, no patients had undergone RVP and one patient (1/35, 2.9%) had a new retinal break after gas injection. Two patients (2/35, 5.7%) developed macular epiretinal membranes during the follow-up, suggesting that the air injection yielded a low risk of postoperative complications.

Scleral cryopexy seems to be more commonly used in previous studies [3,4,5,10]. However, we performed laser retinopexy in our patients after air injection because cryopexy may be associated with a higher incidence of PVR, particularly in young patients. In addition, compared with cryopexy, lasers may produce quicker and stronger chorioretinal adhesion [17]. The use of lasers when dealing with multiple or large retinal breaks is also superior. In our inpatient PR treatment, laser photocoagulation could be performed once the retina was attached. It has been reported that successful reattachment could be achieved after as little as 6–8 h of gas injection [13], which is similar to our present results. However, if an expansile gas were used, laser application may be difficult because the gas bubble would start to expand and might impair the visualization of the retinal breaks. The single PR success rate reached 87.7% (34/39) when our procedure was used, whereas other studies reported single PR success rates of 54% and 59.4% when long-acting expansile gas was used [18,19]. In our study, the air bubbles did not expand and laser treatment was possible as early as 6 h after air injection, which was gradually finished within 3 days for our patients. Therefore, retinal breaks were blocked as soon as possible, which may be one of the reasons for the high retinal attachment rate of single PR treatment in the current study. The other reason for the high success rate may be that we performed PR treatment as an inpatient procedure, which helped us to ensure that the patients maintained the head position properly after air injection and that the position could be adjusted in a timely manner according to the status of the detached retina. In addition, postoperative complications could be noticed during hospitalization; for example, in the case of new retinal breaks after air injection, the break was treated in a timely manner, which prevented further retinal detachment.

The initial PR failed to achieve retinal attachment in four cases in our study. One of them had a missed retinal break or new break, which is suggested to be the most common cause of reoperation following PR [10]. The other failure in our series was a patient with subretinal gas following air injection, and a vitrectomy with internal drainage was performed to achieve retinal attachment. For the remaining two cases of retinal unclosed break, one patient had long axial length and very high myopia of −16 diopter; we believe the relatively unhealthy PRE pump function in the pathologic myopia patient may be the reason for persistent subretinal fluid, and vitrectomy was performed successfully to flatten the retina in this patient. In the other case, subretinal fluid shifted counterclockwise three clock hours after 8 h of head position; the fluid shift induced detachment of the inferior retina, which also had lattice degeneration before PR. The patient did not maintain proper head position at the beginning. The amount of subretinal fluid remained unchanged after 24 h of position. This failed case was further treated successfully with scleral buckle in the two days after PR surgery. Notably, because the PR affords no permanent relief of vitreoretinal traction on the break, the substantial traction after surgery may also be the reason for these two cases of PR failure.

Our study did not exclude patients without a posterior vitreous detachment. Four young patients with no posterior vitreous detachment and atrophic hole were included, and they were all successfully treated by PR in this study. Previous studies have shown the anatomic reattachment rate in pediatric RRD patients undergoing PR was similar to that reported in adult patients. Chen et al. [20] demonstrated the effectiveness of PR for retinal RRD repair in teenagers, and 84.2% of the cases were successfully reattached after one PR treatment. Additionally, Figueiredo et al. [21] found a similar reattachment rate of 85% in 20 pediatric patients undergoing PR treatment for RRD fulfilling PIVOT criteria. The younger patients may not appear to be at higher risk of failing PR. These specific cases usually have no posterior vitreous detachment and healthy RPE pump function; subretinal fluid may resolve more quickly after gas injection. Chen et al. [20] also suggested there may be less new break formation in young patients without PVD. In the condition of no PVD, the gas bubble may pose even and little traction on other parts of the retina and little chance of new break formation. 

PR treatment may result in better postoperative visual acuity than other RRD repair surgeries. Tornambe et al. [22] reported that more patients achieved BCVA ≥20/40 in the PR group (81/92, 88%) versus those in the SB group (57/77, 74%). Moreover, a recent randomized controlled trial demonstrated that the visual acuity outcomes in patients undergoing PR were superior to those undergoing PPV at every follow-up timepoint. The proportion of eyes achieving ≥20/40 was 90.3% compared to 75.3% in the PPV group [3]. Yee et al. [16] showed that long-term visual outcomes following air PR showed comparable improvement with those of PR using long-lasting gases, and 87% of the patients had the same or better vision at the last follow-up. In line with previous studies, no participant had worse postoperative visual acuity in our study, and visual improvement was achieved in 76.2% (16/21) of macula-on RRD and 100% (14/14) of macula-off RRD patients.

The macular structure can be altered in RRD patients even after repair surgeries. Purtskhvanidze et al. [23] and Maqsood et al. [24] found that central retinal thickness was significantly decreased in comparison to that of controls after PPV surgery in RRD patients, especially in macula-off RRD patients. The detached retina may suffer from hypoxic conditions, the outer retinal layers may be deprived of nutrients by the subretinal fluid, and the microvasculature of the inner retinal layers may have lower blood perfusion. Nonetheless, few studies have investigated the changes in retinal structure in RRD patients after PR surgery. Kaderli et al. [25] reported that the inner retinal thickness in the eyes of macula-off RRD patients was significantly higher than that in macula-on RRD patients at 1 month and 3 months post operation. In contrast, we found significantly thinner central retinal thickness in RRD eyes than that in fellow eyes in macula-off patients. This discrepancy in results between these studies may be due to the difference in surgical techniques, follow-up time, and patient populations. We have not investigated the postoperative photoreceptor integrity on OCT images, but it has been suggested that PR treatment may be associated with less discontinuity of the ellipsoid zone and external limiting membrane at 12 months post operation, which may be the reason for the improved visual outcome after PR surgery [26].

Our study presented several limitations. The sample size was relatively small and we included only phakic eyes in the study. A future prospective randomized study with a larger number of eyes and a longer follow-up is needed to verify our findings. We have not compared our PR treatment with other RRD repair surgery treatments such as PPV or SB. A comparative prospective study comparing the outcomes of the two-step procedure of air PR with PPV and SB may yield more conclusive results.

## 5. Conclusions

The inpatient two-step pneumatic retinopexy with pure air injection and early laser photocoagulation may be an effective approach to treating patients with RRD, who presented a high single-operation success rate and good visual acuity recovery.

## Figures and Tables

**Figure 1 jpm-13-00328-f001:**
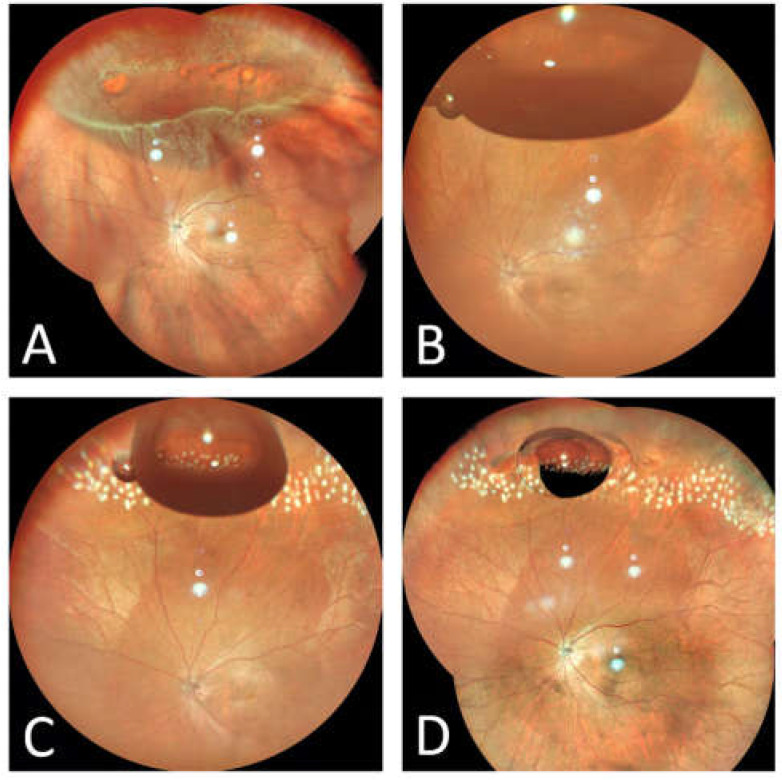
Fundus photography of a typical RRD case before and after PR treatment. (**A**). The patient had superior retinal detachment with multiple breaks. (**B**). The retina has completely reattached at 12 h after intravitreal air injection. (**C**) Laser retinopexy was applied the day after intravitreal air injection. (**D**) Laser retinopexy could be strengthened when the air bubble became smaller.

**Figure 2 jpm-13-00328-f002:**
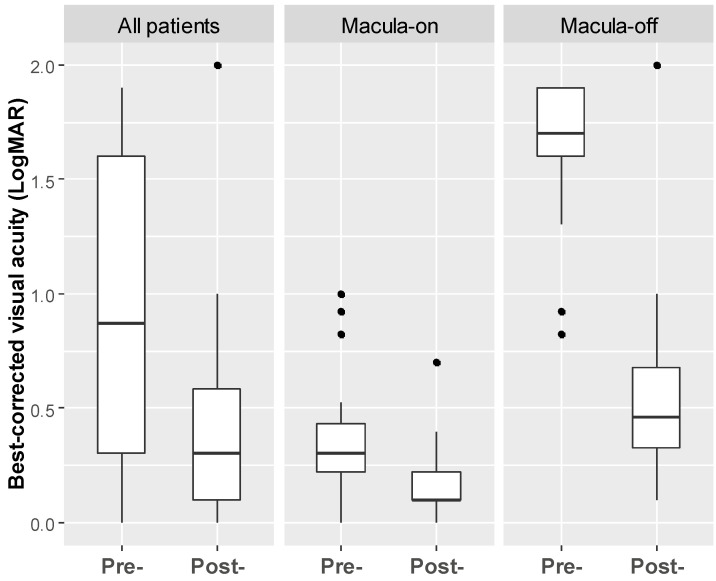
Box-and-whisker plot showing preoperative and postoperative best-corrected visual acuity (LogMAR) in RRD patients undergoing our two-step PR treatment.

**Table 1 jpm-13-00328-t001:** Preoperative clinical characteristics of uncomplicated RRD patients.

Subject Characteristics	Total Patients	Macula-on Patients	Macula-off Patients	*p* ^†^
No. of cases	39	21	18	-
Age, years	51.8 ± 14.5	52.6 ± 10.9	50.8 ± 18.2	0.715
Female, no. (%)	19 (48.7)	11 (52.4)	8 (44.4)	0.652
Right study eye, no. (%)	21 (53.8)	11 (52.4)	10 (55.6)	0.835
Spherical equivalent refractive error, diopter	−3.80 ± 3.75	−4.44 ± 4.59	−3.31 ± 3.14	0.420
Preoperative BCVA, LogMAR *	0.94 ± 0.69	0.36 ± 0.28	1.58 ± 0.36	<0.001
Days to PR surgery ^¶^	12.4 ± 10.0	12.6 ± 10.3	12.3 ± 9.8	0.912
No. of breaks in detached retina	1.28 ± 0.60	1.14 ± 0.48	1.44 ± 0.70	0.106
No. of quadrants of detached retina	1.4 ± 0.50	1.17 ± 0.33	1.67 ± 0.54	0.001

* Counting fingers and hand motion: visual acuity was converted to 1.6 and 1.9 LogMAR, respectively; ^¶^ days to PR surgery was defined as the duration from the day of symptom onset to the day of air injection; † comparisons between macula-on and -off groups; BCVA—best corrected visual acuity; PR—pneumatic retinopexy.

## Data Availability

Not applicable.

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
