# Peer review of "Pneumatic Retinopexy Involving the Use of Intravitreal Air Injection and Laser Photocoagulation for Rhegmatogenous Retinal Detachment in Phakic Eyes"

_jpm, 2023, doi:10.3390/jpm13020328_

Round 1

Reviewer 1 Report

1. More details about break size and air volume. Explain how y define.

2. What were the size and aspects of the breaks? (horseshoe? hole? operculum? meridional tear?

3. about refraction (you measure) but it is not described and possible relations discussed.

4. Add "phakic" to the title.

Reviewer 2 Report

Please find attached a review report in pdf. 

Reviewer 3 Report

Recently, pneumatic retinopexy has been a topic of treatment for retinal detachment. The Authors reported the inpatient two-step pneumatic retinopexy with pure air injection and early laser photocoagulation, resulting in a high single-operation success rate and good visual acuity recovery.
This paper will be very interesting and beneficial for retinal specialists and patients.
I have several questions.

1, In table 1, what does the “Days to PR surgery” mean? Does this mean the number of days between symptom onset and PR treatment? If so, an average of 12 days seemed too long. And “PR surgery” means the day of air injection or laser treatment? The definition of “Days to PR surgery” must be clear.

2, How old was the youngest patient in your study? Don’t your criteria include the condition of PVD? I want to know if your patient includes young patients with an atrophic hole with no PVD. If you excluded such a case, it should be added to the exclusion criteria.

3, Line 163-164; The macula-off group also showed significant improvement in postoperative BCVA (1.62 ± 0.36) of logMAR over baseline (0.59 ± 0.48, P < 0.001, n = 14).

Are the preoperative and postoperative visual acuity values reversed?

4, Line 149-150, what was the reason for “retinal break unblocking (2 cases)”?

 Readers would like to know the reason for this. It would be good if you could comment on this in the discussion.

Round 2

Reviewer 2 Report

Dear Authors, 

Thank you for submitting the corrections. 

Sincerely, 

Reviewer